# Combating Antimicrobial Resistance in the Post-Genomic Era: Rapid Antibiotic Discovery

**DOI:** 10.3390/molecules28104183

**Published:** 2023-05-19

**Authors:** Yuehan Yang, Mara Grace C. Kessler, Maria Raquel Marchán-Rivadeneira, Yong Han

**Affiliations:** 1Translational Biomedical Sciences Program, Ohio University, Athens, OH 45701, USA; yy506416@ohio.edu (Y.Y.); marchan@ohio.edu (M.R.M.-R.); 2Edison Biotechnology Institute, Ohio University, Athens, OH 45701, USA; mk209219@ohio.edu; 3Honors Tutorial College, Ohio University, Athens, OH 45701, USA; 4Department of Biological Sciences, Ohio University, Athens, OH 45701, USA; 5Center for Research on Health in Latinamerica (CISeAL)-Biological Science Department, Pontificia Universidad Católica del Ecuador (PUCE), Quito 170143, Ecuador; 6Department of Chemistry and Biochemistry, Ohio University, Athens, OH 45701, USA

**Keywords:** antimicrobial resistance, genome mining, natural products, antibiotics, biosynthesis gene clusters

## Abstract

Constantly evolving drug-resistant “superbugs” have caused an urgent demand for novel antimicrobial agents. Natural products and their analogs have been a prolific source of antimicrobial agents, even though a high rediscovery rate and less targeted research has made the field challenging in the pre-genomic era. With recent advancements in technology, natural product research is gaining new life. Genome mining has allowed for more targeted excavation of biosynthetic potential from natural sources that was previously overlooked. Researchers use bioinformatic algorithms to rapidly identify and predict antimicrobial candidates by studying the genome before even entering the lab. In addition, synthetic biology and advanced analytical instruments enable the accelerated identification of novel antibiotics with distinct structures. Here, we reviewed the literature for noteworthy examples of novel antimicrobial agents discovered through various methodologies, highlighting the candidates with potent effectiveness against antimicrobial-resistant pathogens.

## 1. Introduction

Antimicrobial-resistant pathogens have always been a serious threat associated with high morbidity and mortality. Multidrug-resistant “superbugs” and emerging new pathogens seem to have brought back the “pre-antibiotic era”, leaving healthcare providers with limited clinical treatment options [1]. Since the first discovery and application of antibiotics in the middle of the last century, resistance has been observed, which may predate antibiotics by millions of years [2]. Microbes use many resistance mechanisms to protect themselves from antibiotics’ cytotoxic or cytostatic effects. Resistant microbes are naturally selected for exposure to common antibiotics that inhibit or kill sensitive counterparts. Moreover, due to lack of competition, the resistant strains can easily grow, spread, and become “superbugs”, such as methicillin-resistant *Staphylococcus aureus* (MRSA) and multidrug-resistant *Mycobacterium tuberculosis*, which often respond poorly to existing medications. 

The Centers for Disease Control and Prevention report over 2.8 million antimicrobial-resistant infections in the United States each year, accounting for more than 35,000 deaths [3]. In addition, the threat of antimicrobial-resistant infections has gotten worse. The COVID-19 pandemic resulted in increased antibiotic use, increased hospitalization-related infections, and decreased progress against antibiotic-resistant infections in the U.S. [4]. Meanwhile, sharp declines in the introduction of novel antibiotics and an associated decline in pharmaceutical industry investment have exacerbated the resistance threat [2]. Antibiotics with novel structures and new modes of action have been and will always be urgently needed. 

Natural products (NPs) and their derivatives have historically served as a prolific source of therapeutics, specifically antibiotics, throughout the history of medicine. After the golden age of natural product discovery from the 1940s to 1950s, the pharmaceutical industry mostly favored structural modifications and semi-synthetic analogs of existing compounds to generate new antibiotics [5]. Figure 1 illustrates the structural evolution of three generations of polyketide antibiotics based on erythromycin. Erythromycin (**1**) is a type I polyketide antibiotic produced by *Saccharopolyspora erythraea* with a 14-membered macrolactone ring and two glucose-derived deoxysugar moieties. The semisynthetic derivatives, clarithromycin (**2**) (methylation at the 6-hydroxy group and 14-membered lactone ring same with erythromycin) and azithromycin (**3**) (15-membered ring and 9-carbonyl group replaced with a methyl-substituted nitrogen), show an enhanced spectrum of activity and are active against erythromycin-resistant *Streptococcus pneumoniae*. However, such efforts far from meet the challenges of antimicrobial-resistant and emerging pathogens. Fortunately, new NP discovery efforts continue to flourish with promising results. In particular, recent genomic and bioinformatic-guided approaches have profoundly accelerated the process of novel antimicrobial agent discovery and overcome the pitfall of a high rediscovery rate through the traditional approaches [6,7]. With the advances in genomics uncovering silent gene clusters, researchers can activate silent gene clusters and obtain the encoded compounds through various techniques. Here, we reviewed the literature from 2002 to 2022 and discussed new compounds against antimicrobial-resistant pathogens discovered by genome mining of natural sources and beyond. In addition, this review examined the antimicrobial mode of action and mechanism of resistance to commonly used therapeutics.

## 2. Genome Mining for Key Enzymes in Biosynthesis Gene Clusters

Genome mining is a bioinformatic approach used to discover and analyze the biosynthesis genetic potential of natural products in the genome of organisms, such as plants, bacteria, and fungi. The most common genome mining approach is to apply computational algorithms to analyze biosynthetic gene clusters (BGCs), targeting known enzymatic or gene functions, particularly core biosynthesis enzymes and specific structural unit biosynthesis enzymes. The underlining rationale is that natural products are assembled, tailored, and regulated by strikingly conservative BGCs. These BGCs are responsible for composing production lines in natural organisms such as microbes, fungi, and plants [8]. 

A typical roadmap of genome mining is elucidated in Figure 1, which represents the discovery of a novel antibiotic through the excavation of bacterial genomes. The natural compound producers were collected from various sources and genomes were sequenced and analyzed for BGCs. Then, researchers used computational algorithms to predict the enzymatic modules and post-assembly tailoring enzymes, as exemplified by the discovery of the novel antibiotic ECO-0501 [9]. The producer strain *Amycolatopsis orientalis* ATCC 43491 was known to be a vancomycin-producing strain. A whole genome analysis indicated it harbors over ten BGCs besides vancomycin. Next, researchers expressed and characterized one of the clusters. A fermentation and chromatography-based separation yielded purified compound(s). Subsequently, structure elucidation through mass spectrometry (MS) and nuclear magnetic resonance spectroscopy (NMR) revealed the structure of a new glycosidic polyketide ECO-0501(**4**). ECO-0501 exhibits strong antibacterial activity against MRSA (MIC: 4.8 μM) and vancomycin-resistant *Enterococci* (VRE) (MIC: 10 μM) [9]. 

BGCs encode enzymes that are responsible for the synthesis of several classes of compounds, including polyketides, nonribosomal peptides, ribosomally synthesized and post-translationally modified peptides, phenazines, and terpenes, among others. Next, we discuss the discovery of novel antimicrobial compounds from the aforementioned classes through various genome-mining approaches. 

### 2.1. Polyketides

Polyketides are biosynthesized by polyketide synthases (PKSs) using acyl-CoA as a precursor. Generally, bacterial type I PKSs have multiple enzymatic modules, each module only adds one unit of acyl-CoA to elongate the polyketide chain, and subsequent catalysis often leads to the formation of the macrocycle (e.g., erythromycin (**1**) in Figure 2). In contrast, bacterial type II PKSs act iteratively, with one enzymatic module being used repeatedly to grow the polyketide chain. Subsequent steps often lead to the formation of a polycyclic aromatic skeleton (e.g., tetracenomycin (**5**) in Figure 2) [10]. The understanding of the biosynthetic rationale behind polyketides has come a long way; using this extensive information, guidelines for predicting cryptic polyketide structures and activity from genome sequences have been developed [11]. 

Besides the example of type I polyketide ECO-0501 we discussed above, additional compounds were discovered through a similar approach. Anthracimycin (**6**), a promising type I polyketide antibiotic with a unique structure, produced by marine-derived *Streptomyces* sp. CNH365, has shown potent activity against anthrax-causing Bacillus anthracis UM23C1–1 (MIC: 0.079 μM) and various antimicrobial-resistance pathogens, such as MRSA (MIC ≥ 0.63 μM) and vancomycin-resistant *S. aureus* (MIC ≥ 0.63 μM) [12]. Anthracimycin has been shown to protect animals from MRSA-induced mortality in a murine model of infection, and macromolecular synthesis studies suggest that it may inhibit RNA and/or DNA synthesis of bacteria [13]. Similarly, the PKS gene cluster in the genome of the myxobacterium *Pyxidicoccus fallax* was investigated, and genes encoding pentapeptide repeat proteins, known to confer resistance to topoisomerase inhibitors, were found in the cluster. Its activation in the natural host and heterologous expression allowed the discovery of pyxidicyclines (e.g., pyxidicyclin A (**7**)) and precursor anthraquinones [14]. 

The heterologous expression approach is often combined with the resistance gene screening approach. Using these two combined approaches, a PKS–NRPS hybrid gene cluster led to the discovery of a series of thiotetronic acid natural products, including the previously known fatty acid synthase inhibitor thiolactomycin (**8**), which has moderate activity against MRSA, *Mycobacterium tuberculosis*, *Plasmodium falciparum*, and *Francisella tularensis* [15]. In a subsequent study, the researchers synthesized structural analogs of thiolactomycin which showed enhanced activity (e.g., TLM 9 (**9**) showed a 75-fold improvement against MRSA) (Bommineni, 2016). A polyketide with a unique tetracyclic skeleton, isoindolinomycin (**10**), has a BGC which is mutated in the wild-type *Streptomyces* sp. SoC090715LN-16. Through screening of rifampicin-resistant mutants and activation of the biosynthetic gene cluster, isoindolinomycin was discovered and characterized [16]. Tetracyclic-type antibiotics, which inhibit bacterial growth by specifically binding to bacterial ribosomal subunits, currently face efflux pump-type resistance due to overuse [17]. Isoindolinomycin may offer new target binding sites and overcome the current antimicrobial resistance issue.

Other than examining the whole PKS clusters, another common methodology is to screen for key biosynthesize units. The synthase genes for 3-amino-5-hydroxybenzoic acid (AHBA), the starter unit in ansamycin-type antibiotics, were screened in a pool of actinobacteria [18]. The genome of *Streptomyces* sp. LZ35, an AHBA synthase-positive strain, was further analyzed for novel BGCs. Multiple attempts were then made to successfully activate the silent neoansamycin BGC through transcriptional regulation factors, leading to the discovery of neoansamycins (**11**, **12**) [19,20]. 

Sometimes silent BGCs can be activated through adjustment of the microbial culture conditions. Keyicin (**13**), a poly-nitroglycosylated anthracycline antibiotic, was discovered by co-culturing two marine bacteria from the otherwise silent PKS gene cluster in the producing strain *Micromonospora* sp. In addition to its possible interspecies crosstalk function, keyicin exhibits selective antimicrobial activity against Gram-positive bacteria including *Rhodococcus* sp. and *Mycobacterium* sp. and it does not cause nucleic acid damage [21]. 

Moreover, genomics leveraging analytical techniques (e.g., LC-MS, HPLC, NMR, CryoEM, etc.) [22,23,24,25,26] has been used to profile the metabolic phenotypes of microbes for prospecting antibiotics. A discovery platform was built that applies genomics, metabolomics, and antimicrobial activity screening of LC-MS-based metabolomic arrays from 174 marine animal isolated bacteria [27]. *Micromonospora* sp. WMMC-415’s hidden chemical diversity was highlighted and prioritized by its potent activity against *Candida albicans* and led to the discovery of turbinmicin (**14**), a broad-spectrum antifungal drug candidate that targets multidrug-resistant fungi with a high safety profile and a highly selective mechanism of action [27]. 

Genome mining of BGCs is often combined with activity-guided separation. Clostrubin (**15**), a highly unsaturated aromatic polyketide from an obligate anaerobe *Clostridium beijerinckii*, demonstrated pronounced activity against various pathogenic Gram-positive bacteria, including MRSA (MIC: 0.12 µM), VRE (MIC: 0.97 µM), and mycobacteria (MIC: 0.48 μM), and moderate antiproliferative and cytotoxic effects [28]. In another study, clostrubin and its analog (**16**) were found to be produced by *Clostridium puniceum* functioning as antibiotics against other plant pathogens, giving the anaerobic bacteria an advantage to compete in an oxygen-rich plant environment [29]. Clostrubins are an exemplary case of mining the genomic biosynthesis potential in combination with traditional culture condition screening and/or molecular biology tools, leading to the rapid discovery of polyketide antibiotics. 

### 2.2. Nonribosomal Peptides

Distinguished from peptides and proteins synthesized from the ribosomal pathway, nonribosomal peptides are assembled by modular machinery consisting of large, selective multifunctional enzymes called nonribosomal peptide synthetases (NRPS). The products of these machineries are many successfully marketed drugs (e.g., penicillin (**18**), vancomycin (**19**), and daptomycin (**20**) shown in Figure 3) [30]. Over the last few decades, the complicated production process of nonribosomal peptides has been the subject of biochemical and structural biology investigations, offering new compounds with pharmaceutical value. 

Straightforwardly, conserved genes of BGCs can be screened to mine new nonribosomal peptides with pharmaceutical value. Twenty-seven marine actinomycete isolates were subjects for NRPS and PKS gene screening, and coupled with a bioactivity assay, this led to the identification of *Nocardiopsis* sp. TFS65-07. Further isolation and examination determined the active compound to be TP-1161 (**21**), a new macrocyclic thiopeptide with three thiazole and two oxazole groups. TP-1161 displayed superior activity against a panel of Gram-positive pathogens including VRE (MIC: 0.84 μM), and the mode of action for thiopeptide is yet to be determined [31]. Similarly, genome mining of NRPS gene clusters in *Brevibacillus laterosporus* DSM 25 led to the discovery of Brevibacillin 2V (**22**), which has a strong antimicrobial activity against difficult-to-treat antimicrobial-resistant Gram-positive pathogens with low hemolytic activities and cytotoxicities to eukaryotic cells. When combined with marketed antibiotics, Brevibacillin 2V demonstrates a synergistic effect against Gram-negative pathogens and shows a high stability in human plasma, most likely because of the inclusion of D-amino acids and non-canonical amino acids [32]. Likewise, a cyclic lipopeptide medipeptin A (**23**) was isolated from the microbiome of a tomato plant in the Netherlands. Researchers took a sample from the plant and isolated *Pseudomonas* bacteria, the genomic DNA of which was then analyzed for biosynthetic gene clusters. Medipeptin A was isolated and showed good activity against multiple pathogens. Medipeptin A’s antibiotic capabilities come from its binding with lipoteichoic acids and by forming pores in the bacterial membrane [33]. 

Many nonribosomal peptides encoded by unculturable bacteria genomes can be obtained in the heterologous expression hosts. Malacidins (**24**, **25**), which are calcium-dependent nonribosomal lipopeptides, target the cell wall biosynthesis of multidrug-resistant pathogens. The malacidin gene cluster was captured from desert soil metagenomics using primers targeting calcium-binding motifs. To avoid rediscovery, a phylogeny tree of the calcium-dependent antibiotic gene was generated to guide the selection of a novel malacidin gene cluster. Heterologous expression yielded new antibiotic malacidins, whose activity spectrum against multiple drug-resistant pathogens was evaluated and the mode of action was discussed [34]. Similarly, taromycin A (**26**) and B (**27**) are nonribosomal lipopeptides whose BGCs highly resemble the daptomycin gene cluster. The 67 kb otherwise silent taromycin BGC was directly cloned and refactored into the model expression host *Streptomyces coelicolor*. Taromycins show moderate activity against multiple daptomycin-resistant strains, demonstrating an alternative option for a new antibiotic [35,36]. Although the heterologous expression of NRPS clusters often proceeds with difficulty due to the giant gene cluster size (30–100 kb), the success of malacidins and taromycins demonstrated new avenues for antibiotic discovery. 

Another breakthrough method to excavate NPs from unculturable bacteria was to grow them directly in their native environments, such as soil and seawater. Teixobactin (**28**) was discovered using a device that allows source bacteria to grow between semipermeable membranes immersed in the original soil sample, enabling nutrients, and signaling molecules to promote growth. Teixobactin represents a new class of antibiotics with excellent activity against Gram-positive pathogens and a non-detectable resistance [37]. In addition, new antibiotics may come from previously overlooked sources such as the animal microbiome. Lugdunin (**29**) is a thiazolidine-containing nonribosomal peptide that is produced by the human microbiota *Staphylococcus lugdunensis*. Ludunin displays a broad spectrum of activity against Gram-positive pathogens and prevents *S. aureus* from colonizing. Researchers also observed that *S. lugdunensis* nasal colonization was linked to a notably low *S. aureus* carriage rate, indicating that lugdunin or lugdunin-producing commensal bacteria may prevent staphylococcal infections [38].

NP bioinformation-guided and organically synthesized peptides are a new approach to rapidly obtaining antimicrobial agents. Macolacin (**30**), an analog of the nonribosomal peptide antibiotic colistin (**31**), is a synthetic bioinformatic natural product generated by chemical synthesis from the BGC-guided approach. Through an extensive search of bacteria genomes for colistin-like BGCs, a BGC from *Paenibacillus xylanexedens* mac was projected to produce a structurally divergent colistin congener. Subsequential chemical syntheses produced macolacin and its derivatives, which exhibit superior activity over colistin against multidrug-resistant pathogens, including drug-resistant *Acinetobacter baumannii* and colistin-resistant *Neisseria gonorrhoeae* [39]. A similar chemical synthesis approach was applied to produce humimycins, which are MRSA active antibiotics, from human-associated bacteria NRPS clusters. Two NRPS clusters from the genomes of *Rhodococcus equi* and *Rhodococcus erythropolis* predicted the amino acid sequences of humimycins A (**31**) and B (**32**), respectively [40]. Using synthetic endeavors, more nonribosomal peptides (Paenimucillins (**34**–**36**)) and an antifungal peptide (**37**) were discovered. Paenimucillin C (**36**) inhibits multidrug-resistant *Acinetobacter baumannii* clinical isolates and sterilizes multidrug-resistant *A. baumannii* infection with no indication of a rebound in a murine model [41,42]. 

The synthetic bioinformatic NP approach bypasses complicated gene cluster activation in the original host, avoids establishing a heterologous expression system for large gene fragments, skips the time-consuming fermentation extraction and isolation, and alleviates the difficulties of complicated peptide structure elucidation. However, synthetic peptides usually cannot generate the final products of the BGCs, partly due to the accuracy of the computational prediction of versatile and complicated post-assembly tailoring enzymes. Despite the limitations, this approach offers adequate peptide skeletons for a structure–activity relationship exploration of the analogs. Among post-assembly tailoring enzymes, cytochrome P450s successfully generated activity-improved nonribosomal peptides. Inspired by the significant antibacterial property enhancement of the dimeric products, Guo et al. searched Streptomyces genomes for tailoring enzyme cytochrome P450s with the function of dimerizing structural similar nonribosomal peptides [43]. The overexpression of two cytochrome P450s in *Streptomyces alboflavus* sp. 313 led to the generation of dimeric nonribosomal peptide di-alboflavusins. Di-alboflavusin A_1_ (**38**) and A_2_ (**39**) demonstrate a 10–100-fold improvement in antibacterial activity against multiple MRSA strains compared with the counterpart monomers (marked green in Figure 3) [43]. Figure 3 shows the nonribosomal peptides discussed in this section. 

### 2.3. Ribosomally Synthesized and Post-Translationally Modified Peptides

Ribosomally synthesized and post-translationally modified peptides (RiPPs) present high complexity and miscellaneous bioactivities for pharmacological applications [44]. The intricate RiPP molecular structures arise from diverse peptide skeletons and complex post-assembly modifications, generating linear peptides, cyclic peptides, lanthipeptides, lasso peptides, thiopeptides, lipopeptides, glycopeptides, etc., presenting many unique drug candidates for antimicrobial-resistant infections [45]. 

Many RiPPs are found through simple genome mining for BGCs. Lactocillin (**40**), a thiopeptide antibiotic, is a classic example of a product of massive genome data mining of bacterial BGCs from unconventional biological systems. Lactocillin BGC from a vaginal microbiota genome was pinpointed out of 3118 BGCs from human-associated bacteria genomes. Lactocillin displayed potent activity against vaginal pathogens. Additionally, metatranscriptomic sequencing data show that lactocillin is expressed in the human vagina in vivo, suggesting vaginal microbiota may inhibit pathogenic bacteria and prevent vaginal infections through the expression of lactocillin-type thiopeptides [46]. 

The novel two-peptide lantibiotic lichenicidin (**41**) was isolated from Bacillus licheniformis D13 and was found using NCBI-BLAST to look through the B. licheniformis genome to detect gene clusters that could produce novel lantibiotics. Lantibiotics are a class of RiPP antibiotics characterized by post-translational addition of the thioether amino acids lanthionine and methyllanthionine [47]. Lichenicidin has activities against Bacillus megaterium, Bacillus subtilis, *S. aureus*, Micrococcus luteus, and Rhodococcus spp. [48]. Another RiPP found through genome mining of BGCs is thiomuracin (**42**). Thiomuracin is a thiopeptide found by genome mining of the bacterium *Thermobispora bispora.* The peptide has strong activities against methicillin-resistant *Staphylococcus aureus* USA300 (MIC of 0.18 μM) and vancomycin-resistant *Enterococcus faecium* U503 (MIC of 0.046 μM). The peptide also has an MIC of 1.5 μM against spore-forming Bacillus anthracis str. Sterne. Thiomuracin is post-translationally modified into a cyclic structure that has inhibitory effects on bacterial protein synthesis [49]. 

Other novel RiPPs were found through targeted genome mining, where an analog or type of gene cluster was searched for. Specialicin (**43**) is a lasso peptide that was found by searching the Streptomyces specialis genome for analogs of known peptides. A BLAST search was run, looking for BGCs that were similar to those of siamycin. Specialicin was then found, isolated, and tested, which revealed an MIC of 3.75 μM against M. luteus [50]. Another lasso peptide, Lassomycin (**44**), showed promising activities against *Mycobacterium tuberculosis* (0.41–1.65 μM), including many resistant strains (0.21–1.65 μM). Lassomycin was found in an extract of *Lentzia kentuckyensis*. It is an interesting lasso peptide because its N-terminus tail does not string through its macrolactam ring similar to many other lasso peptides [51]. 

Promoter engineering and synthetic modular regulatory elements have successfully turned on silent BGCs for new NP production and constructed overproducing strains of known antibiotics [52]. New bottromycin derivatives with 5–50-fold higher titers of bottromycin A_2_ (**45**) were generated by replacing native promoters with randomized constitutive synthetic promoters. The expression of the bottromycin gene cluster is decoupled from intricate native regulatory networks via promoter engineering and heterologous expression. Additionally, this tactic enhanced the production of bottromycins, a class of macrocyclic peptides with antimicrobial activities against multidrug-resistant bacteria [53,54]. 

Some RiPPs are heterologously expressed in a different host than the gene they are found in, one such compound is citrocin (**46**). Citrocin is an antimicrobial lasso peptide found in both *Citrobacter pasteurii* and *Citrobacter braakii*. It was heterologously expressed in Escherichia coli and then isolated. Its structure was determined, and MIC values were calculated for its antimicrobial activities against a handful of species. Citrocin displayed an MIC ranging from 16–100 μM for *E. coli* species with 16 μM for enterohemorrhagic *E. coli* O157:H7 strain TUV93-0. The compound also displayed an MIC of 1000 μM against *Salmonella enterica* serovar Newport and 125 μM against a clinical isolate of Citrobacter [52]. An additional compound, venezeulin (**47**), was found by genome mining for lantibiotic synthetases [55]. Venezeulin, isolated from *Streptomyces venezuelae*, is heterologously produced in *E. coli* BL21 [56]. 

Similar to synthetic bioinformatic nonribosomal peptides, RiPPs can be synthetically produced. TL19 (**48**) is an antibiotic peptide (MIC of 0.9 to 15 μM against E. faecium) made by synthetically combining the N-terminus portion of nisin and the C-terminus portion of haloduracin α (HalA1 is one of the two peptides that make up the lanthipeptide haloduracin) [56]. Nisin, an FDA-approved safe peptide food preservative with potential for clinical use, is a well-studied 34-residue lantibiotic that is produced by *Lactococcus lactis*, where its synthesis is regulated by a two-component regulatory system. Haloduracin is a two-peptide antibiotic. Both HalA1 and Nisin contain a lipid II binding site [57]. The aforementioned representative RiPP antibiotics are shown in Figure 4. 

### 2.4. Phenazines

The fundamental structure of phenazines is composed of redox-active nitrogen tricyclic aromatic rings; however, they differ in the modification groups. Many phenazines have broad spectrum antimicrobial properties, making them prospective antibiotics [58,59,60]. The biosynthesis of the phenazine core structure (marked pink in Figure 5), which diverges from the shikimic acid pathway, has been well studied. Bacterial producers have a conserved set of phenazine biosynthesis genes (PHZ) that encode enzymes that convert chorismic acid into complex phenazine precursors phenazine-1-carboxylic acid (PCA) (**49**) or phenazine-1,6-dicarboxylic acid (PDC) (**50**) [61]. Similar tactics have been employed to unearth new phenazines to those used for other classes of antibiotics discussed previously. 

Phenazine BGC screening and bioactivity-guided isolation are two commonly combined approaches to excavate new phenazine antibiotics. A bioinformatics analysis of Kitasatospora sp. HKI 714 genome revealed phenazine BGC and the subsequent bioactivity-guided isolation revealed new terpenoid-substituted phenazines endophenazine A1 (**51**), F (**52**), and G (**53**). Heine et al. analyzed the biological effects of the new endophenazines, mostly concentrating on their antibacterial properties against various mycobacteria and MRSA [62]. Our group applied the bioinformatic tool antiSMASH 6.0 [63] to analyze the genome of *Amycolatopsis nigrescens* DSM 44992, revealing one phenazine BGC as well as thirty other BGCs. An antimicrobial-activity-guided isolation revealed a novel phenazine compound YYH001 (**54**), whose absolute structure was determined by crystallography. However, additional bioactivity assays did not find appreciable antibiotic properties of the purified YYH001. We speculated that the superior activity against bacterial and fungal pathogens was brought by other BGC products of the crude extract. In fact, we successfully identified a family of novel polyketide antibiotics from a PKS gene cluster in *Amycolatopsis nigrescens*. Besides the assessment of BGCs, researchers often use universal primers of conserved genes to find the microbes’ genetic potential for antibiotic production. This approach is still suitable for cost-effective and time-efficient screening of large numbers of microorganisms. We used universal gene phzE primers to evaluate a pool of actinobacteria that our group isolated from unusual biosamples. The positive strain *Streptomyces* sp. B23 isolated from a soil sample was selected for further investigation. A bioactivity-guided separation led to the discovery of a novel antibiotic kephenamycine (**55**), along with several known phenazines (e.g., saphenamycin (**56**)). 

The diverse bioactivity of phenazines arises from a variety of modifications to the core. Shi et al. reported that the entomopathogenic bacterium *Xenorhabdus szentirmaii* underwent multiple modifications of the phenazine core through two different BGCs, generating derivatives with phenazine–polyketide hybrid scaffolds (e.g., phenaszenketide A–C (**57**–**59**)) and phenazine–peptide hybrid scaffolds (e.g., pelagiomicin B (**60**) and phenaszentine C (**61**)). The two BGC pathways were connected via a common aldehyde intermediate altered by several enzymatic and non-enzymatic reactions. The evaluation of the antibiotic activity of the generated phenazine derivatives points to a highly efficient method for converting the phenazines with specialized activity against Gram-positive bacteria into broad spectrum antibiotics [64]. However, the antibiotic activity of phenazines against pathogens was evaluated via diameters of inhibition zones in disc diffusion assays and lack of replications; MICs against antimicrobial-resistant pathogens need to be conducted to comprehensively assess their antibiotic properties. Other research discussed the antibacterial mode of action of a structurally similar phenazine–peptide _D_-alanylgriseoluteic acid (**62**), which induced the SOS response through the endogenous redox activity of the compound [65]. Other types of phenazine antibiotic modification seen in nature include glycosylation (e.g., endophenazine B (**63**) and C (**64**)) [66], dimerization (e.g., esmeraldin B (**65**) and diastaphenazine (**66**), two cores marked pink in Figure 5) [67,68], and N-oxidation (e.g., myxin (**67**) and iodinin (**68**)) [69], among others. 

Activation of silent gene clusters poses one of the major challenges of genome mining for new antibiotics. Low doses of antibiotics may act as signaling molecules that can activate or modulate BGCs. Streptophenazines A–H (**69**–**76**), a family of new phenazines, were found in a marine Streptomyces isolate. The production pattern of streptophenazines depends on the different antibiotics added to the culture conditions which act as signaling molecules. Specifically, tetracycline induces streptophenazine F and G production and increases the yield of streptophenazine A–D, whereas bacitracin induces streptophenazine H production. The streptophenazines showed moderate activity against Bacillus subtilis as well as Staphylococcus lentus [70]. Heterologous expression and promoter engineering may also activate silent phenazine BGCs. A remarkable chemical diversity of streptophenazines was produced by heterologous expression and refactoring promoters of a PHZ/PKS/NRPS hybrid gene cluster from *Streptomyces* sp. CNB-09. Among the 112 isolated compounds, a streptophenazine analog containing an unprecedented N-formylglycine moiety (marked in blue in Figure 5), streptophenazine Q (**77**), demonstrated an effective antibiotic activity against MRSA and disease-causing group A *Streptococcus*. However, the heterologous expression of phenazine BGCs can be challenging due to the antibiotic effect of the products on the heterologous host without a proper resistance mechanism [71]. The endophenazine BGC in the original host (Streptomyces anulatus) produces a much higher yield of antibiotic endophenazine A (**78**) than the heterologous host Streptomyces coelicolor M512. The authors found that the heterologous host strains accumulated a glutamine adduct (marked in blue in Figure 5) endophenazine E (**79**), which does not have antibiotic activity, and they speculated that glutamination could be a defense mechanism of the host [72,73]. 

Furthermore, NP-inspired organic synthesized phenazines exhibit extraordinary pharmaceutical value against antimicrobial-resistant pathogens and eradicate persistent pathogen biofilms. Clofazimine (**80**) has been applied to treat multidrug-resistant Mycobacteria tuberculosis, may compete with an essential component in the mycobacterial electron transfer chain, and releases reactive oxygen species [74]. Clofazimine derivatives (**81** and **82**), which displayed a remarkable potency improvement in vitro and reduced lipophilic-associated side effects, were further examined for pharmacokinetic profiles in vivo [75,76,77]. Coelho et al. examined the in vitro activity of several phenazine derivatives against rifampicin-resistant *M. tuberculosis* and pan-susceptible *M. tuberculosis* H37Rv and found that the allyl-pyran-group-containing phenazine (**83**) showed an MIC of 2.2 μM and the bromo-pyran-group-containing phenazine (**84**) showed an MIC of 16 μM [78]. More excitingly, marine brominated phenazine (**85**) inspired the synthesis of a series of halogenated phenazines (e.g., **86**–**87**) whose structure–activity relationships were thoroughly investigated by Prof. Huigens and colleagues. Potent halogenated phenazine analogs successfully eradicated biofilms formed by persistent pathogens through rapid induction of iron starvation with minimal mammalian cytotoxicity and hemolysis. These findings informed and encouraged the design of rational prodrugs for pharmacological translation [79,80,81,82]. The structures of the aforementioned representative phenazines are shown in Figure 5. 

### 2.5. Terpenes/Terpenoids

Terpenes are often thought to be the largest class of natural products. They often come from plants but are also common in fungi and bacteria. Terpenes are made of repeating isoprene units and typically have carbons in multiples of five because of the five-carbon nature of isoprenes [83]. Specifically, terpenoids are compounds made of isoprene units that may contain functional groups such as oxygen, while terpenes are hydrocarbons composed only of hydrogen and carbon atoms, but these terms are often used interchangeably in the literature [84]. 

A BGC analysis can also indicate the genetic potential of producing terpene-type antimicrobials. Researchers sequenced the genome of actinobacteria *Streptomyces* spp. NRRL S-4 and searched for pentalenolactone analogs using the antiSMASH platform. The results revealed six terpene BCGs. Two were determined to produce two sesquiterpenoids 1-deoxy-8α-hydroxypentalenic acid (**88**) and 1-deoxy-9β-hydroxy-11-oxopentalenic acid (**89**). These two compounds had MICs of 64 and 61 μM, respectively, against *S. aureus* ATCC 25923 and 128 and 61 μM against *E. coli* ATCC 25922, respectively [85]. Another terpenoid found through genome mining is benditerpenoic acid (**90**). The BGC of benditerpenoic acid, a novel bicyclic diterpenoid, was found in the genome of *Streptomyces* sp. CL12-4. The compound was then heterologously expressed in *E. coli*, isolated, and tested to find its MIC against *E. faecium* (401 μM), *S. aureus* (200 μM), MRSA (200 μM), multidrug-resistant *S. aureus* (401 μM), and *B. subtilis* (100 μM) [86].

More often, terpenoids are found through the traditional approach of bioactivity screening of extracts or broths. *Sarcophyton trocheliophorum,* a species of soft coral found in the Red Sea, has been found to produce five terpenoid compounds. Two of these compounds are novel and pyrane-based and were exemplified by sarcotrocheliol (**91**). Sarcotrocheliol showed substantial antibiotic activity with MICs of 1.53 μM against *S. aureus* and 3.06 μM against MRSA. Cembrene-C (**92**) was also isolated from this coral and had impressive antifungal activities, with an MIC of 0.68 μM against *Candida albicans* and *Aspergillus flavus* [87]. A marine sponge of the genus *Hippospongia* found in Fiji produced the sesquiterpenoid quinone epi-ilimaquinone (**93**). This natural product had antibiotic activities of 182 μM against MRSA, 91 μM against *S. aureus*, 45 μM against VRE, and 364 μM against amphotericin-resistant *Candida albicans* [88]. 

Among many terpenes and terpenoids, one class of compounds has very distinct chemical properties. The diterpene sordarin (**94**), a glycoside antibiotic with a tetracyclic diterpene core, was first discovered in the 1970s and isolated from *Sordaria araneosa*. Sordarin has been studied increasingly since its discovery to allow for the discovery of more, similar antimicrobial compounds [89,90]. The antifungal agent targets elongation factor II and halts protein synthesis. Sordarin is the subject of many studies now, with the aim of finding new analogs through genome mining and synthesis. Kudo et al. studied the genome sequence of *Sodaria araneosa* Cain ATCC 36386 to reveal the sordarin biosynthesis pathway. Understanding the biosynthesis of sordarin allows for the targeted synthesis of analogs. FR290581 (**95**), a sordarin analog, was synthesized by modification of sordaricin. It has an enhanced activity (MIC of 0.781–1.56 μM) against *Candida* species [90]. Another analog, GR193663 (**96**), was synthesized by fusing the sugar moiety, creating a new innovative structure that had increased antifungal activities. The new compound GR193663 showed superior activity (MICs at the nanomolar level) against *Candida* species [91]. The structures of the aforementioned terepenes/terpenoids are shown in Figure 6. Table 1 summarizes the compounds with activity against antimicrobial-resistant pathogens discussed in the above five sections. 

## 3. Mechanisms of Resistance and Promising Drug Leads

The discovery of novel compounds through various technologies offers great promise for combating infectious diseases. However, it is equally important to understand the mechanisms of resistance that may arise to develop strategies to minimize the emergence of resistant strains and prolong the effectiveness of these compounds. In this context, it is essential to identify compounds that can serve as promising drug leads for the development of new antibiotics and antifungal drugs. Apart from their potency, promising drug leads should also exhibit favorable pharmacological properties such as a high selectivity and minimal toxicity. In this section, we will discuss some of the compounds that meet these criteria. 

### 3.1. Combating Antimicrobial-Resistant Bacteria

Resistance mechanisms are emerging and currently exist for all currently available antibiotics [94]. The common resistance mechanisms are summarized in Figure 7. Among the most prescribed drugs, β-lactam antibiotics are characterized by the four-membered cyclic amide ring (e.g., penicillin in Figure 3). β-lactam antibiotics covalently bind to penicillin-binding proteins, which are enzymes involved in the last stages of peptidoglycan cross-linking in bacteria and inhibit the synthesis of bacterial cell walls [95]. There are at least two common bacterial resistance mechanisms to β-lactam antibiotics: (1) producing β-lactamase, an enzyme that breaks the β-lactam ring structure and (2) producing altered penicillin-binding proteins with a decreased affinity for most β-lactam antibiotics [96]. Another common type of antibiotic are macrolides, which contain a macrocyclic lactone. Represented by erythromycin (**1**), macrolides impede the bacterial protein synthesis by binding to the 50S ribosomal subunit. Resistance mechanisms of macrolides include (1) upregulating efflux pump expression levels to remove intracellular macrolides, (2) mutating bacterial ribosome proteins and ribosomal RNA, and (3) reducing the binding affinity of the macrolides by phosphorylating the sugar moiety or hydrolyzing the macrocyclic lactone [97]. In addition, broad spectrum antibiotic rifampin represents the ansamycin-type of antibiotics. Rifampin is a frontline treatment for tuberculosis, whose bactericidal properties stem from high-affinity binding to and inactivating the bacterial DNA-dependent RNA polymerase [98]. Goldstein argued that rifampin resistance is caused by a genetic modification in the bacterial RNA polymerase β subunit [99], whereas Javid et al. concluded that phenotypic resistance to rifampicin is caused by the mistranslation of mycobacterial proteins, suggesting that mistakes in protein translation might affect how bacteria respond to stress [100]. Resistance to conventional antibiotics has caused polymyxins to be reexamined as viable treatment choices. Polymyxins represented by colistin (30) are considered the last resort for multi-drug-resistant Gram-negative pathogens. Colistin exerts a bactericidal effect by binding to the lipid A moiety of lipopolysaccharide, an important Gram-negative bacteria outer surface component, leading to rupture of the cell membrane [39]. Even though colistin resistance is infrequent due to its limited usage, several strains of colistin-resistant clinical isolates have recently been found. Other than the common antibiotic resistance mechanisms, researchers have speculated that lipopolysaccharide modification may contribute to colistin resistance [101]. Additional antibiotic examples and molecular targets are summarized in Figure 8. 

The following four antibiotic candidates possess the superior traits of structural innovation and/or unique modes of action that deserve more attention and resources for advanced research of clinical translation. Macolacin (**30**) is an analog of colistin, the discovery of which by genome mining we briefly discussed previously. It is important to examine the mode of action for macolacin to overcome colistin resistance. Wang et al. postulated the distinct structure of macolacin may improve the ability to interact with the lipid A moiety of lipopolysaccharide for colistin-resistant strains. Teixobactin (**28**) inhibits the synthesis of the bacterial cell wall by binding to a highly conserved motif of lipid II and lipid III, both of which are precursors of the bacteria cell wall components (peptidoglycan and teichoic acid, respectively) [39]. The most outstanding finding is that researchers did not detect acquired resistance after passaging *S. aureus* and *M. tuberculosis* for 25 days with a sub-MIC level of teixobactin, this finding implies the natural resistance to teixobactin may take years to develop. The above two candidates both belong to the nonribosomal peptide category, whereas the next candidate is a ribosomally synthesized peptide with an interesting loop and tail structure. Lassomycin (**44**) targets ATP-dependent protease and kills *M*. *tuberculosis* with a high specificity while offering no hemolysis and low cytotoxicity towards mammalian cells. Without the clear presence of persistence, lassomycin demonstrated a stronger bactericidal effect against stationary *M*. *tuberculosis* than rifampin [51]. It appears to have a new antimicrobial mode of action by blocking the protein degradation pathway of mycobacteria. Lastly, halogenated phenazines (**83–85**) stand out because of their antibiofilm capacity. Unlike most phenazine compounds’ redox activity-related bactericidal effects, halogenated phenazines bind to metal (II) cations, leading to rapid downstream iron starvation and biofilm eradication. Halogenated phenazine prodrugs were designed based on structure–activity relationships with potent effectiveness (minimal biofilm eradication concentration: 0.59–18.8 μM) and low toxicity (HeLa cytotox >100 μM and RBC lysis 2.7% at 200 μM), giving the compounds a wide margin of safety [82]. 

### 3.2. Combating Antimicrobial-Resistant Fungi

Antimicrobial-resistant fungal infections are more concerning due to their high motility and the lack of treatment options. Burgeoning organ transplants and use of medical implants, prolonged hospitalization, and the HIV/AIDS and COVID-19 pandemics have dramatically increased the prevalence of life-threatening invasive fungal infections [103,104,105,106]. Currently, only three classes of antifungal drugs are approved in the clinical practice guidelines for invasive fungal infections and heavily rely on targeting ergosterol in cell membranes and glucans in cell walls [107,108,109,110]. The over-reliance on limited molecular targets has led to the rapid emergence of antifungal drug resistance. 

The molecular targets of current marketed antifungal drugs and promising new drug leads are elucidated in Figure 9. Frontline treatments (azoles), exemplified by itraconazole, are synthetic antibiotics that inhibit lanosterol 14-α-demethylase and interrupt the synthesis of ergosterol, resulting in the accumulation of toxic sterol intermediates and membrane stress [111]. Azole resistance mechanisms are developed by (1) altering 14-α-demethylase to decrease the binding affinity, (2) overexpressing efflux pump ABC transporters, and (3) altering sterol biosynthesis to incorporate 14-α-methyl fecosterol in the cell membrane instead of ergosterol [108]. Another frontline prescription for a systemic fungal infection is amphotericin B, a polyene macrolide antibiotic produced by the type I polyketide synthetase (PKS) of *Streptomyces nodosus* [112]. The most accepted fungicidal mode of action is the pore creation model, in which polyenes bind to ergosterol to create an ion-channel-like complex and leak intracellular contents from the fungal cells. Other study models propose that amphotericin B acts similar to a hydrophobic “sponge” and specifically extracts ergosterol from the fungal membrane. Given the mode of action of polyenes is heavily dependent on ergosterol, resistance mechanisms have been developed by changing the fungal membrane sterol composition. Additionally, tolerance to oxidative stress may improve the fungi’s ability to withstand exposure to amphotericin B [113]. Considered as the last resort for antimicrobial-resistant or invasive fungal infections, echinocandins (e.g., echinocandins B) belong to the nonribosomal lipopeptide class produced by filamentous fungi. Echinocandins block the synthesis of 1,3-β-D-glucan, an important component of the fungal wall anatomy, causing significant stress and disruption of the cell wall integrity [114,115]. Mutations in FKS genes, which encode molecular targets of the enzyme 1,3-β-D-glucan synthase, are predominantly responsible for echinocandin resistance in *Candida* species [108]. 

The following drug leads with new antifungal targets merit special attention for pharmaceutical development. Firstly, sordarins (e.g., **94**), a class of tetracyclic diterpene antibiotics first discovered in 1969, are obtained from various natural sources, mostly from different fungi, and have varied structural components. By inhibiting the activity of fungal ribosomes and elongation factor 2, both of which are absent in mammalian cells, sordarins specifically impede the production of fungal proteins. However, no obvious candidates have been reported for clinical development and clinical trials [116,117]. Secondly, heat-stable antifungal factor (HSAF) (**17** in Figure 2), a tetramic-acid-containing macrolactam encoded by a PKS-NRPS hybrid gene cluster from *Lysobacter enzymogenes* C3, demonstrates strong activity in a wide range of fungi [118]. The antifungal mode of action and biosynthesis regulation factors were later thoroughly investigated by Prof. Du’s lab; they found that HSAF inhibits the production of fungal sphingolipids by specifically targeting ceramide synthases of filamentous fungi which do not present in mammals and plants. Additionally, HSAF induced ROS-medicated apoptosis in *Candida albicans* and significantly reduced the fungal burden in a murine model [119,120]. Thirdly, turbinmicin (**14**), which we discussed previously as a great candidate from the polyketide class for multidrug-resistant fungal infection. Mode-of-action studies suggested that turbinmicin targets fungal-specific Sec14 of the vesicular trafficking pathway, in contrast to other antifungal drugs. In addition to its potent efficacy in vitro and in vivo, the low toxicity and well-tolerant dosage of turbinmicin pave the way for its imminent preclinical study and translational development [27].

## 4. Conclusions

This article reviews the literature from 2002 to 2022, focusing on five classes of antimicrobial agents discovered from natural sources or inspired by natural products. We examined the mechanisms of antimicrobial resistance (AMR) and reviewed the modes of action of promising antimicrobial drug leads. Moreover, we emphatically discussed the studies where various genome mining approaches were combined with either traditional natural product research technologies or state-of-the-art multidisciplinary technologies. 

Although natural products exhibit a high degree of diverse stereochemistry, their biosynthetic enzymes are highly conserved and can be used to screen biosynthetic strains and their corresponding types of natural product compounds. Nowadays, genomic sequencing generates massive data and has become an integral part of modern biotechnological research. Its rapid growth has led to the development of bioinformatics tools that can analyze this information. The accumulative biosynthetic pathway research of numerous known compounds has assisted the development of algorithms to predict the enzymatic assembly lines (e.g., PKS and NRPS) and the structures of NPs derived from gene sequences. In addition to searching for NP biosynthesis enzymes, other mining approaches include (1) phylogeny-based excavation, (2) regulator screening, (3) tailored enzyme screening, (4) untargeted metabolomics, etc. Moreover, synthetic biology, molecular biology, and advanced analytical technologies are often integrated into today’s natural product discovery strategies. These new approaches have armed researchers with the necessary tools for speedy and efficient novel antibiotic discovery, to fight against antimicrobial-resistant infections, and to stay ahead of pathogen evolution. 

In addition to pouring resources into the development of new antibiotics, an in-depth understanding of antibiotic modes of action is needed to prescribe the relevant drug. Combinational therapy using antibiotics with different modes of action will also maximize efficacy. Furthermore, the explicit and detailed regulation of antimicrobial usage in healthcare settings worldwide as well as in the agriculture and animal husbandry industries is important to minimize or slow the spread of antimicrobial resistance.

## Data Availability

Not applicable.

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
