# Peer review of "Combating Antimicrobial Resistance in the Post-Genomic Era: Rapid Antibiotic Discovery"

_molecules, 2023, doi:10.3390/molecules28104183_

Round 1

Reviewer 1 Report

1-      This is not a keyword (mode of action)

2-      The section title should be Conclusion only without prospects.

3-      Conclusion section is long, reduce it with a brief of this article findings.

4-      References style should be consistence, for example recheck reference 12.

Author Response

Point 1: This is not a keyword (mode of action)

Response 1:  “Mode of action” is deleted from keyword list

Point 2: The section title should be Conclusion only without prospects.

Response 2:  Conclusion section is updated.

Point 3: Conclusion section is long, reduce it with a brief of this article findings.

Response 3:  A paragraph of the conclusion section is separated from the manuscript.

Point 4: References style should be consistence, for example recheck reference 12.

Response 4:  Reference list has been updated.

Reviewer 2 Report

The article "Combating Antimicrobial-Resistance in Post-Genomic Era: 2 Rapid Antibiotic Discovery" is a review about the development of novel drugs starting from biosynthetic molecules that are modified and implemented to have a certain effect also on resistant microorganisms. It summarizes the literature of the last 10 years about the topic, with a huge work data search and elaboration. 

The article is well written and interesting for the scientific community, it is understandable and goes straight to the point, without unecessary frills.

I recommend it for the publication on Molecules after few minor corrections:

Layout - in figures, the title should be converted into a caption and the note integrated on it, as suggested in the Molecules sample text. The same should be done for schemes and tables. Moreover, text should be "justify" and "et al." in Italic. I know I'm picky but this should be done, at least for aestethics. It's a nice article, but the great content could take only advantage from the "form", if you know what I mean. Hope to not being offensive (but to be of help) suggesting you the Molecule template link: https://www.mdpi.com/files/word-templates/molecules-template.dot

The acronym BGCs is never explained in the text and this could create troubles to young readers or the ones that are not confident with this filed. If I well understood the text, it should be "biosynthetic gene clusters", it would be nice if it could be reported once just the first time that the acronym is found in the text (line 85). The same for TS (line 495)

Images in figure 3, 4 and 5 are quite small. May I ask to enlarge them or make a selection of some of them to improve the size, please?

Figure 6 lacks of the comment. It could be enough if you change the title into a caption.

Table 1 is too large and doesn't fit the page. My advise is to change it in order to be conform with the rest of the article. 

Since the sudden change of topic at line 697, may I suggest to entitle the new paragraph as "conclusion"? I think it would the best way to conclude the article and saperate the last two paragraphs from the rest of the manuscript.

References - some of them have the first letter of the title in Capital while other not. Also some doi are absent while other present. Please aligne everything in just one style. (I know I know I'm picky)

Anyway, great job. 

Author Response

Point 1: Layout - in figures, the title should be converted into a caption and the note integrated on it, as suggested in the Molecules sample text. The same should be done for schemes and tables. Moreover, text should be "justify" and "et al." in Italic. I know I'm picky but this should be done, at least for aestethics. It's a nice article, but the great content could take only advantage from the "form", if you know what I mean. Hope to not being offensive (but to be of help) suggesting you the Molecule template link: https://www.mdpi.com/files/word-templates/molecules-template.dot

Response 1:  The figures and schemes captions have been updated. “et.al” have been italicized.

Point 2: The acronym BGCs is never explained in the text and this could create troubles to young readers or the ones that are not confident with this filed. If I well understood the text, it should be "biosynthetic gene clusters", it would be nice if it could be reported once just the first time that the acronym is found in the text (line 85). The same for TS (line 495)

Response 2:  Acronym BGC was explained when first time show up in line 83. "TS" was deleted, and the sentence was rewritten in line 495 to increase clarity.

Point 3: Images in figure 3, 4 and 5 are quite small. May I ask to enlarge them or make a selection of some of them to improve the size, please?

Response 3:  Fig. 3,4,5 were enlarged and reorganized to better show the structures.

Point 4: Figure 6 lacks of the comment. It could be enough if you change the title into a caption.

Response 4:  Fig.6 comment is updated, and the title is changed into a caption.

Point 5: Table 1 is too large and doesn't fit the page. My advise is to change it in order to be conform with the rest of the article.

Response 5:  Table 1 has been updated according to the Molecule template.

Point 6: Since the sudden change of topic at line 697, may I suggest to entitle the new paragraph as "conclusion"? I think it would the best way to conclude the article and saperate the last two paragraphs from the rest of the manuscript.

Response 6:  One paragraph of the conclusion is deleted to reduce the length and keep the topic consistent.  Conclusion section title is updated.

Point 7: References - some of them have the first letter of the title in Capital while other not. Also some doi are absent while other present. Please aligne everything in just one style. (I know I know I'm picky)

Response 7:  Reference list has been updated.

Reviewer 3 Report

The article is well-written, informative, and comprehensive. It provides a clear overview of the current state of the art in natural product discovery for antimicrobial applications. It also acknowledges the existing knowledge's limitations and gaps and suggests further research directions.

My only minor concern is that the authors must define and explain genome mining and its methods. I think a better description of Scheme 1 would benefit the reader.

Author Response

Point 1: My only minor concern is that the authors must define and explain genome mining and its methods.

Response 1:  Genome mining was defined in line 80.

Point 2: I think a better description of Scheme 1 would benefit the reader.

Response 2:  Scheme 1 description was updated.